# The Function of Bed Management in Pandemic Times—A Case Study of Reaction Time and Bed Reconversion

**DOI:** 10.3390/ijerph20126179

**Published:** 2023-06-19

**Authors:** Chiara Barchielli, Milena Vainieri, Chiara Seghieri, Eleonora Salutini, Paolo Zoppi

**Affiliations:** 1Management and Healthcare Laboratory, Institute of Management, Sant’Anna School of Advanced Studies, 56127 Pisa, Italy; milena.vainieri@santannapisa.it (M.V.); chiara.seghieri@santannapisa.it (C.S.); 2Department of Nursing and Obstetrics, Azienda USL Toscana Centro, 50123 Firenze, Italypaolo.zoppi@uslcentro.toscana.it (P.Z.)

**Keywords:** COVID pandemic, bed management, system solidness

## Abstract

The last decade was characterized by the reduction in hospital beds throughout Europe. When facing the COVID pandemic, this has been an issue of major importance as hospitals were seriously overloaded with an unexpected growth in demand. The dichotomy formed by the scarcity of beds and the need for acute care was handled by the Bed Management (BM) function. This case study explores how BM was able to help the solidness of the healthcare system, managing hospital beds at best and recruiting others in different settings as intermediate care in a large Local Health Authority (LHA) in central Italy. Administrative data show how the provision of appropriate care was achieved by recruiting approximately 500 beds belonging to private healthcare facilities affiliated with the regional healthcare system and exercising the best BM function. The ability of the system to absorb the extra demand caused by COVID was made possible by using intermediate care beds, which were allowed to stretch the logistic boundaries of the hospitals, and by the promptness of Bed Management in converting beds into COVID beds and reconverting them, and by the timely management of internal patient logistics, thus creating space according to the healthcare demands.

## 1. Introduction

Bed Management (BM) is a form of proactive resource control based on the constant assessment of hospital services and incoming patient flow from the Emergency Department (ED) toward the most appropriate setting for care [1]. Boaden and colleagues [2] defined it as the process of reconciliation between the demand and the supply of beds, which is a scarce resource. BM is generally carried out by a dedicated team that provides real-time operational status of the capacity of the hospital to receive. The BM team is usually formed by nurses [3], and many hospitals around the world adopted various forms of electronic systems to help with the necessity of having real-time bed status availability [4]. A recent systematic review [5] underlines that the appropriateness of this function depends on a variety of uncertain factors, which may take the form of a patient length of stay (LoS), fluctuations in healthcare demand, and unexpected admissions, just to name a few. The complex nature of BM classifies it as part of the Operations Management (OM), a variety of managerial practices that design and control the production processes and the production of services. It is referred to as the way in which the organization creates the highest level of efficiency possible [6]. BM is a function that must be deeply rooted in the hospital structure as it must consider reality constraints, namely the number of available beds and making them available with the appropriateness based on the patient’s condition. These functions generate data or routinely collected health data that were the basis for this paper. The latter analyzes the timeliness and ability of the BM team’s operations in a large LHA in central Italy during the COVID outbreak. This unparalleled calamity provided the opportunity to observe the organizational behavior of the healthcare system during a public health crisis that presented not only clinical but also managerial criticalities. The first two waves of the spreading infection took place in the time frame from the 15 March 2020 to the 15 May 2020 and the second from the 15 September to the 15 December 2020. The performed analyzes are aimed at depicting the commitment to providing care for the COVID-infected and non-infected patients, creating a “bed-buffer” through the recruitment of approximately 500 beds from the intermediate care setting. As context information, pre-pandemic USLTC BM function can be described as a tool that was used to move and allocate patients on clinically based logic: the present clinical condition was the criteria on which the appropriate setting inside the hospital was determined. The pandemic BM function quickly took the form of a coordination that acted as a network, not only with the whole hospital system, but with the community care facilities. We can define it as the control room that acted against the bottlenecks and the frequent unforeseen events that we will be describing.

### An Italian Case Study

The 2019 OECD Report “Health at a Glance” [7] shows how Italy is characterized by a low number of hospital beds: 2.6 beds per 1000 inhabitants. This is the result of policies aimed at reducing acute care beds, in line with the general European trend: since 2000 in all EU countries, the number of beds per capita fell by 20% [8]. The Italian National Healthcare System (NHS) is regionally based, the State holds the power to address and control the regional policies and outcomes, and public care is largely free of charge. The 20 regions are free to organize the provision of care in the way they estimate to be the best to meet their residents’ needs. In 2015 a reorganization of the hospital care network was undertaken and alternative forms of hospitalization were created (e.g., community care facilities and intermediate care facilities) alongside a binding hospital planning criterion: the equipment of hospital beds must not exceed 3.7 beds per 1000 inhabitants.

This case study focuses on the USL Toscana Centro (USLTC), an LHA with a catchment area of 1,700,000 inhabitants on a surface of 5000 square kilometers (that coincides with the medium-sized cities of Firenze, Prato, Pistoia and Empoli) in which 13 hospitals are based. This LHA integrates different services that extend from the prevention of illnesses to long-term care. Table 1 summarizes its activity during 2020:

To cope with the COVID emergency, the Italian Ministry of Health increased the number of inpatients and Intensive Care Unit (ICU) beds, the latter being a worldwide utter concern [9]. At the end of the summer of 2019, other interventions were put in place to reinforce both community care and intermediate care with the aim of relieving the pressure on hospitals, although with a different capacity throughout the Italian regions [10]. Pecoraro et al. [11] describe how the robustness of the Italian (and Spanish) hospitals’ structural components was much smaller than the German and French, which could count on a larger number of beds, which mitigated the effects of the massive request for hospitalization at the beginning of the pandemic. The Spanish case studies [12,13] reported their adopted strategies to face the outbreaks and the initial shock, contributing to the enrichment of the knowledge of the lessons learned. Even if the hospitals’ robustness was smaller, those organizations put in place interventions (e.g., the use of indicators to catch the dynamicity of the organizations’ responses, the modeling of bed occupancy, etc.) that were able to exceed the structural limits of the hospitals.

The starting point was not the best but, as presented in the recent literature [11], Tuscany reported positive results in bed management during the COVID pandemic, and this is attributed to the ability to handle complex cases determining a short length of stay (LoS) when compared to the national average.

This case study aims to describe the monthly trends of COVID and no-COVID bed use during the first two waves of the pandemic and the interval in three care settings: (i) the ICU and sub-acute ICU (sub-ICU, a step-down from the intensive care setting which still provides more intensive care than the inpatients setting), (ii) the inpatient (internal medicine, surgery), and (iii) the intermediate care, through the use of administrative healthcare data.

Concerning intermediate care, it has to be said that they are meant to be a place for the most chronically ill, elderly patients, a middle level between acute hospital care and basic care. During the pandemic, they were used as a buffer, a way to alleviate the pressure on hospitals, and it was crucial to their functioning [13,14].

## 2. Materials and Methods

Before getting to the core of the paper, a clarification about the way beds are counted when occupied by an infected patient is necessary, as reported in Figure 1:

In a room with non-infected patients, beds are all available and computed as physically employable. If the room hosts an infected patient, as in the case of COVID infection, not all the beds are available, hence computable, even if vacant. This is crucial to understand that hospital beds are not a flexible, or at least a little fungible, resource. In the eventuality of a droplet-transmitted virus, as the Coronavirus, a droplet isolation protocol has to be put in place [15,16] and this implies having less space for beds, the number of which is reduced. The intrinsic case study design [17] was adopted as a means to shed light on a unique phenomenon, the effects of the pandemic in a particular organization. Data for the analysis were gathered from the administrative data flows. As Benchimol and colleagues [18] illustrated in their work on routinely collected healthcare data in the absence of a previous research aim, it was necessary to create a methodological instrument that could address the growing availability of this type of information in the attempt to use it to improve not only clinical but management decisions as well. The aim of the authors was to “develop a reporting guideline for observational studies using health data collected for non-research purposes as an extension of STROBE—the REporting of studies Conducted using Observational Routinely-collected Data (RECORD) statement” [18]. The transparency in reporting observational case studies, such as this one, was needed to address the potential research biases. The RECORD checklist presents 13 items to be followed to correctly use this kind of data.

In particular, the consulted information flows in our work were the Hospital Discharge Record flow, the Intermediate Care Occupancy flow and the administrative data used to populate CROSS (Centrale Remota per le Operazioni di Soccorso Sanitario, Remote Central for Health Rescue Operations) in the aforementioned periods of time, identified as the first pandemic wave, the second pandemic wave and the interval period. The CROSS system is a patient placement tool: regions can activate it when no beds are available in their territories and ask other regions for help. Its basic requirement is the ability to ensure, at least for the first 72 h after the emergency hospitalization of the patient, adequate space availability, technological equipment and human resources in order to offer the patient the best care possible. Data were extracted from a database that the Bed Manager created ad hoc and from the various interfaces used in the LHA.

We first explored the number of hospitalized patients and the mortality rates among the different settings. Finally, we calculated both the monthly number of available beds and the Beds Occupancy Rate (BOR), namely the occupied inpatient beds as a % of the available beds over the observation period (first wave, interval and second wave), in the three settings of interest to investigate if the BM was able to remove the bottleneck created by intensive care units (ICU) and sub-ICUs through the fast conversion and reconversion of beds from no-COVID to COVID (and vice versa). Although there is no consensus over the ideal BOR percentage, 85% might be considered an optimal threshold to reduce the risk of bed shortages [16,19].

The BOR parameter has taken on a particular importance during the pandemic, stimulating the creation of tools to explore and track different countries’ decisions. As an example, the Oxford COVID-19 Government Response Tracker (OxCGRT) is a database that tracks and shows the governments’ responses in relation to changes in the pandemic patterns, deciding policies such as lockdowns, contact tracing activities and various types of containment in relation to the parameters from the BOR [17,20].

## 3. Results

During the entire period, 7098 patients were admitted. A total of 742 died (10.4%), 46 (4%) of which died in the intermediate care setting as shown in Table 2. As expected, the highest in-hospital mortality rate was among ICU-sub-ICU patients (29%).

During the observation period, 88 patients were assessed in the Emergency Department (ED) and sent directly to intermediate care. Table 3 resumes the breakdown of the hospital re-admission of the 88 patients that were directly sent to the intermediate care setting when diagnosed.

The re-admission of 88 patients was necessary because patients faced subtle deterioration of their clinical conditions after it appeared manageable, if not resolved. In Table 4, the total monthly COVID and no-COVID beds and BORs are reported for the first wave, the interval (after the national lockdown) and the second wave. As a side note, this data collection began ten days after the pandemic outbreak in Tuscany. Indeed, the system had no idea about the nature of the threat as no objective data were available [21] given that the first Italian case was notified on the 20 February 2020. Healthcare personnel were literally unaware of the kind of response to put in place as nothing was known about the natural history of the infection, namely symptoms, adequate treatments and recovery time.

The BOR has varied throughout the two waves, with a first peak in April, when the intermediate care beds worked as a system outlet valve: they registered a BOR of 83% while the BOR of inpatient beds was at 71%. During the summer months, all the intermediate care beds were reconverted into no-COVID beds to allow the existing no-COVID healthcare demands to be addressed. In the same interval period, the no-COVID inpatients BOR recorded a decline of 23%. During the second wave, the number of hospitalized patients was higher than the one registered in April, as all the beds were saturated: ICU’s and sub-ICU’s BOR was 78%, inpatients’ BOR and intermediate care’s BOR were 90% and 98%, respectively. The ICU and sub-ICU occupancy rates during the first wave was lower because of the higher mortality rate, as the treatments were administered to patients empirically, which is based on a clinical hypothesis yet not supported by complete information. Data show greater use of ICU and sub-ICU beds and a minor use of inpatient beds during the first wave as ICUs and sub-ICUs were extended into operating rooms, transforming them into negative pression-rooms. Intermediate care beds were activated “on demand”: during the summer, there was no need to occupy them. Conversely, in the autumn during the second wave, intermediate care beds were increased to sustain the system in coping with the increase in the healthcare demand. In Figure 2, the total availability and total occupancy of beds in the entire territory of the USLTC are reported in the period between the 15th of March and the 15th of December: as it shows in the trend of COVID beds, in the first months of the pandemic, a lot of beds were unoccupied. This can be attributed both to the lack of elements of evaluation of the virus behavior and to the total suspension of all elective operating sessions. After the summer interval, the difference between the total beds and occupied beds is strongly reduced, especially from mid-October onwards. This trend demonstrates how the BM function was more than efficient for managing the processes of conversion and re-conversion of beds and how it was able to take full advantage of the additional beds from the intermediate care sector.

For what concerns the trend in No-COVID beds, the delta between available and occupied beds is bigger. This can be this can be and imputed to the slowdown suffered by elective surgery.

## 4. Discussion and Conclusions

The COVID pandemic created several issues of major importance, among which was a serious hospital overload due to an unexpected growth in demand. We investigated the relationship between COVID and no-COVID beds during the first two waves of the pandemic in three settings: inpatients, ICU and sub-ICU and intermediate care. Inpatients and intermediate care beds were characterized by a very high BOR during the first wave, which was even higher in the second wave. The number of admissions was high, as well as the mortality. The ability of the system to absorb the extra demand caused by COVID was made possible using intermediate care beds, which were allowed to stretch the logistic boundaries of the hospitals [20] and by the promptness of Bed Management in converting beds into COVID beds and reconverting them, and by the timely management of internal patient logistic, thus creating space according to the healthcare demands.

The pandemic made it evident that the hospital must be a place for acute patient care and that community care is the key to the sustainability of the healthcare systems as it will be the place to treat and care for chronic patients. Hospitals are operating among tight margins in terms of staff and number of beds [22], and the pandemic shock it represented.

The home must be the first place of care: this is the major political and managerial aim to achieve because it will give flexibility to the whole healthcare system when facing external stresses. For this reason, USLTC is investing in community care solutions, e.g., (i) building multidisciplinary community care teams that can bring their expertise to the patient at home, (ii) investing in the role of the Family and Community Nurse that is responsible for meeting the needs of patients and identifying the latent ones, and (iii) placing resources in the development of intermediate care facilities. Among the lessons learned during this period, we must also mention the importance of having emergency plans for hospitals, in the sense of having designated isolation areas or designated areas that can quickly be converted for this purpose, and the fact that the USLTC BM function still is performing its function as a control room that allocates and moves patient not only on a clinical basis but also on a managerial one by keeping close and frequent contacts between all the nodes of the communication and decision network.

This study presents several limitations: data were not promptly registered, and they refer to a single LHA. We described a system that appeared to be resilient, as the system has responded to pandemic pressures but it is not possible to evaluate its performance because the benchmark is missing. In order to compare the efficiency of the management, especially for what concerns the BM, we should have data related to the same period of other similar companies in size and available staff. What is certain is that the healthcare system that we used to know does not exist anymore, and the changes that countries around the world will have to undergo mainly relate to the strengthening of community care. As Griffin et al. highlight [23], during pandemic times it is important to balance the connection between the hospital and the territory it serves: the hospital can be the first responder but not the only one. Data support the choice to rely on intermediate care beds to relieve pressure from the hospitals and, therefore, treat a larger number of patients [24]. 

## Figures and Tables

**Figure 1 ijerph-20-06179-f001:**
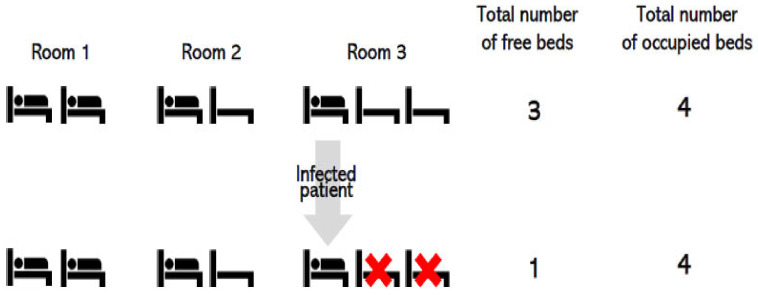
How to compute beds when an infected patient is sharing space with other non-infected patients.

**Figure 2 ijerph-20-06179-f002:**
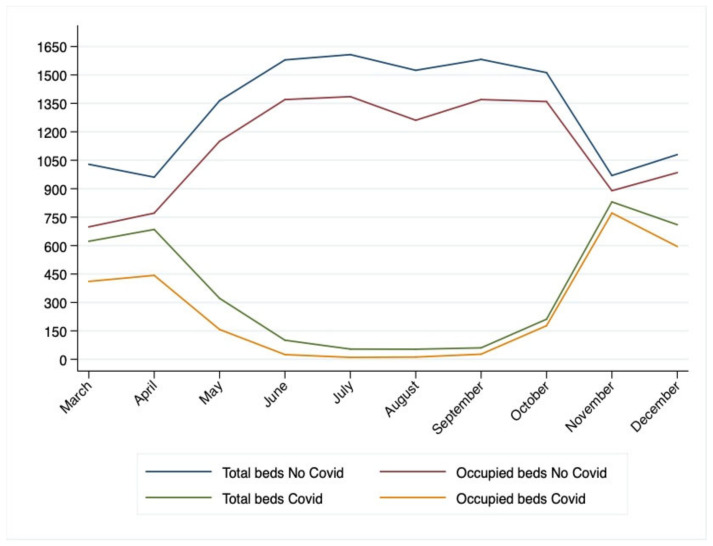
Trends in total COVID and No-COVID beds/occupied beds from the 15th of March to the 15th of December 2020.

**Table 1 ijerph-20-06179-t001:** USLTC, 2020 activities.

Year	Acute Beds	Occupancy Days	ED Accesses	Surgeries	Outpatient Services
2020	2735	668,094	416,566	58,786	22,734,702

**Table 2 ijerph-20-06179-t002:** Total number of assessed and deceased patients 15 March 2020–15 December 2020.

Setting	Admitted	Deceased	In-Hospital Mortality
Intermediate care	1124	46	4%
Inpatient	5470	549	10%
ICU-sub ICU	504	147	29%
Total	7098	742	10%

**Table 3 ijerph-20-06179-t003:** Hospital re-admission of patients initially assigned to the Intermediate care setting.

Month	Hospital COVID Readmissions from Intermediate Care
March	0
April	15
May	26
June	17
July	1
August	0
September	1
October	6
November	5
December	17

**Table 4 ijerph-20-06179-t004:** Number of COVID and No- COVID beds and BORs during the first wave, the interval and the second wave.

Timing	Month	Setting	COVID Total Beds	BOR COVID	No-COVID Total Beds	BOR No-COVID
Wave 1	3-May	Intermediate care	35	35%	0	0%
Wave 1	3-May	Inpatients	449	71%	954	68%
Wave 1	3-May	ICU-sub ICU	146	57%	75	63%
Wave 1	4-April	Intermediate care	90	83%	0	0%
Wave 1	4-April	Inpatients	454	66%	895	81%
Wave 1	4-April	ICU-sub ICU	141	48%	65	67%
Wave 1	5-May	Intermediate care	123	69%	14	64%
Wave 1	5-May	Inpatients	241	50%	1136	88%
Wave 1	5-May	ICU-sub ICU	63	39%	109	70%
Interval	5-May	Intermediate care	94	51%	33	56%
Interval	5-May	Inpatients	86	33%	1321	84%
Interval	5-May	ICU-sub ICU	44	27%	118	75%
Interval	6-June	Intermediate care	41	39%	53	57%
Interval	6-June	Inpatients	41	23%	1383	89%
Interval	6-June	ICU-sub ICU	23	6%	143	75%
Interval	7-July	Intermediate care	0	0%	52	51%
Interval	7-July	Inpatients	32	31%	1408	89%
Interval	7-July	ICU-sub ICU	22	0%	147	71%
Interval	8-August	Intermediate care	0	0%	38	58%
Interval	8-August	Inpatients	31	37%	1340	85%
Interval	8-August	ICU-sub ICU	22	2%	145	67%
Interval	9-September	Intermediate care	0	0%	39	81%
Interval	9-September	Inpatients	34	59%	1378	87%
Interval	9-September	ICU-sub ICU	22	16%	144	72%
Wave 2	9-September	Intermediate care	0	0%	57	65%
Wave 2	9-September	Inpatients	41	63%	1401	90%
Wave 2	9-September	ICU-sub ICU	24	19%	144	76%
Wave 2	10-October	Intermediate care	40	90%	72	80%
Wave 2	10-October	Inpatients	167	90%	1305	92%
Wave 2	10-October	ICU-sub ICU	37	55%	136	78%
Wave 2	11-November	Intermediate care	100	98%	45	99%
Wave 2	11-November	Inpatients	626	93%	828	93%
Wave 2	11-November	ICU-sub ICU	104	86%	96	78%
Wave 2	12-December	Intermediate care	118	94%	45	82%
Wave 2	12-December	Inpatients	493	83%	938	93%
Wave 2	12-December	ICU-sub ICU	100	78%	96	80%

## Data Availability

No new data were created or analyzed in this study. Data sharing is not applicable to this article.

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
