# Peer review of "The Function of Bed Management in Pandemic Times—A Case Study of Reaction Time and Bed Reconversion"

_ijerph, 2023, doi:10.3390/ijerph20126179_

Round 1
Reviewer 1 Report
Dear authors:
I have reviewed the manuscript entitled “The Function of Bed Management in Pandemic Times. A case Study of Reaction Time and Bed Reconversion. Your study explores how bed management was able to help the solidness of the healthcare system, managing hospital beds at best and recruiting others in different settings as intermediate care, in a large Local Health Authority (LHA) in central Italy. Administrative data show how the provision of appropriate care was achieved recruiting approximately 500 beds belonging to private healthcare facilities affiliated with the regional healthcare system and exercising at best the bed management function.
The thematic was a big concern, especial to health management. But I believe your work should reflect more the importance of the outcomes to the actual situation, being your data from 2020, is crucial that you justify and make clear you statement.
I believe you could review the references you used (40% older than five years). You should translate all Italian to English (intermezzo). In discussion I believe you could explore more the actual idea of home hospitalization, because the political measures are investing in that area.
I have nothing to add and I wish you all the best with its publication.
I have nothing to add and I wish you good luck towards publishing the paper!
Author Response
We really thank you for your time and expertise, we do appreciate you found the time to revise and give us advice to make this article better.
The thematic was a big concern, especial to health management. But I believe your work should reflect more the importance of the outcomes to the actual situation, being your data from 2020, is crucial that you justify and make clear you statement. I believe you could review the references you used (40% older than five years). You should translate all Italian to English (intermezzo). In discussion I believe you could explore more the actual idea of home hospitalization, because the political measures are investing in that area.
We added a section with the lessons learnt that can address this concern (lines 52-58: As a context information, pre-pandemic USLTC BM function can be described as a tool that was used to move and allocate patients on a clinically based logic: the present clinical condition was the criteria on which the appropriate setting inside the hospital was determined. The pandemic BM function quickly took the form of a coordination that acted as a network not only with the whole hospital system, but with the community care facilities. We can define it as the control room that acted against the bottlenecks and the frequent unforeseen events that we are going to describe.) that is underlined in the conclusion of the paper as well (lines 239-242: … and the fact that the USLTC BM function still is performing its function as a control room that allocates and moves patient not only on a clinical basis, but also on a managerial one by keeping close and frequent contacts between all the nodes of the communication and decision network.).
We updated the reference list, including recent papers and similar case studies.
The word “intermezzo” was replaced by the word “interval”. Throughout the document.
We also added an explanation of why the political side is investing in the home hospitalization/home as the first place of care: (lines 226-236: The pandemic made it evident that the hospital must be a place for acute patient and that the community care is the key to the sustainability of the healthcare systems as it will be the place to treat and care for chronic patient. Hospitals are operating among tight margins in terms of staff and number of beds [22], and the pandemic shock represented . Home must be the first place of care: this is the major political and managerial aim to achieve, because it will give flexibility to the whole healthcare system when facing external stresses. For this reason, USLTC is investing in community care solutions, e.g., (i) building multidisciplinary community care teams that can bring their expertise to the patient at home, (ii) investing in the role of the Family and Community Nurse that is responsible for meeting the needs of patients and identifying the latent ones, (iii) placing resources in the development of intermediate care facilities.)
Thank you again for your precious feedback.
Reviewer 2 Report
Dear Authors,
Thank you very much for the opportunity to review the manuscript "The Function of Bed Management in Pandemic Times. A Case Study of Reaction Time and Bed Reconversion." This case study was written from experience derived from the USL Toscana Centro (USLTC) during 2020.
The construct of the manuscript is neat and easy to follow. However, I have a few minor constructive feedback, hoping to substantiate the manuscript more effectively.
Introduction: it may be relevant to provide more background information on how bed management and allocation were performed before the pandemic – particularly between 2015 to 2019 (ie, before the pandemic outbreak).
Results: it may be helpful to provide information on the average length of stay between the Covid vs non-Covid patients, particularly by the type of settings, such as intermediate care, inpatient, ICU and Sub-ICU. Information on the LOS from admission to discharge and the ICU LOS if relevant.
Discussion: more in-depth discussion regarding the bed management functions can entice the readers' interest instead of diving almost immediately into the limitation of the case study. I would be very interested to read about the insight/lesson learnt from the pandemic. What will be the future direction referencing to the bed management system in synchrony with the overall initiatives of bed reduction across local Toscana vs EU.
Thanks again for the opportunity to review this interesting article.
The command of English is generally good. No major issue that have to comment.
Author Response
Estimated reviewer, we do thank you for your time and your precious time in formulating advice for our manuscript.
We followed your feedback and provided information on how BM worked before the pandemic, and we focused on the lesson learnt.
Introduction: it may be relevant to provide more background information on how bed management and allocation were performed before the pandemic – particularly between 2015 to 2019 (ie, before the pandemic outbreak).
We addressed this comment in lines 52-58: As a context information, pre-pandemic USLTC BM function can be described as a tool that was used to move and allocate patients on a clinically based logic: the present clinical condition was the criteria on which the appropriate setting inside the hospital was deter-mined. The pandemic BM function quickly took the form of a coordination that acted as a network not only with the whole hospital system, but with the community care facilities. We can define it as the control room that acted against the bottlenecks and the frequent unforeseen events that we are going to describe.
Results: it may be helpful to provide information on the average length of stay between the Covid vs non-Covid patients, particularly by the type of settings, such as intermediate care, inpatient, ICU and Sub-ICU. Information on the LOS from admission to discharge and the ICU LOS if relevant.
This information will be part of an upcoming article, and the data of this very firs period were not reliable, so we preferred to leave them out.
Discussion: more in-depth discussion regarding the bed management functions can entice the readers' interest instead of diving almost immediately into the limitation of the case study. I would be very interested to read about the insight/lesson learnt from the pandemic. What will be the future direction referencing to the bed management system in synchrony with the overall initiatives of bed reduction across local Toscana vs EU.
We addressed this comment in this part: lines 226-242:
The pandemic made it evident that the hospital must be a place for acute patient and that the community care is the key to the sustainability of the healthcare systems as it will be the place to treat and care for chronic patient. Hospitals are operating among tight margins in terms of staff and number of beds [22], and the pandemic shock represented Home must be the first place of care: this is the major political and managerial aim to achieve, because it will give flexibility to the whole healthcare system when facing ex-ternal stresses. For this reason, USLTC is investing in community care solutions, e.g., (i) building multidisciplinary community care teams that can bring their expertise to the pa-tient at home, (ii) investing in the role of the Family and Community Nurse that is respon-sible for meeting the needs of patients and identifying the latent ones, (iii) placing re-sources in the development of intermediate care facilities. Among the lessons learned during this period, we must also mention the importance of having emergency plans for hospitals, in the sense of having designated isolation areas or designated areas that can quickly be converted for this purpose and the fact that the USLTC BM function still is per-forming its function as a control room that allocates and moves patient not only on a clin-ical basis, but also on a managerial one by keeping close and frequent contacts between all the nodes of the communication and decision network.
we do hope this manuscript will receive your favour this time.
Reviewer 3 Report
This is an important manuscript addressing hospital bed management during the C-19 crisis. Overall, the paper is interesting and provides a well-organized overview of the topic. My comments are mostly minor.
1) In general, administrative hospital data is notoriously unreliable. Please address how the data was validated.
2) Ideally, as the authors note, it would have been useful to have comparable data related to the same period of other similar companies in size and available staff. Granted that such data is difficult to obtain, especially for a pandemic time period, historic data from their hospitals would be a second option to serve as a referent.
3) Perhaps the authors can contrast their analysis with other papers in the literature that have addresses similar hospital bed bottlenecks, such as during a natural disaster, war, influenza outbreak, etc.
4) A nice strength of the paper is it simplicity and readability. On the other hand, the manuscript noticeably lacks any inferential statistics and scientific rigor beyond basic tabulation by month. Some graphic visualization would be a plus.
No major issues with quality of English language.
Author Response
Estimated reviewer, we really thank you for your comments and for the time you dedicated to make our work better.
We included in the paper the methodological steps we took (RECORD checklist) to make the use of data worthy. We also included recent papers about case studies that can serve as reference.
We however did not make any inferences on the data we collected as a choice, as that will be the material for another article we are currently working on and that takes into consideration a larger timespan. We did not find the data we considered now as sufficient to infer, being them scarce.
This is an important manuscript addressing hospital bed management during the C-19 crisis. Overall, the paper is interesting and provides a well-organized overview of the topic. My comments are mostly minor.
1) In general, administrative hospital data is notoriously unreliable. Please address how the data was validated.
This point was addressed in this part: lines 118-128: Data for the analysis were gathered from the administrative data flows. As Benchimol and colleagues [18] illustrate in their work on routinely collected healthcare data in absence of a previous research aim, it was necessary to create a methodological instrument that could address the growing availability of this type of information in the attempt to use them to improve not only clinical, but management decisions as well. The aim of the au-thors was to “develop a reporting guideline for observational studies using health data collected for non-research purposes as an extension of STROBE- the REporting of stud-ies Conducted using Observational Routinely-collected Data (RECORD) statement” [18]. Transparency in reporting in observational case studies like this one was needed to ad-dress the potential research biases. The RECORD checklist presents 13 items to be fol-lowed to correctly use this kind of data.
2) Ideally, as the authors note, it would have been useful to have comparable data related to the same period of other similar companies in size and available staff. Granted that such data is difficult to obtain, especially for a pandemic time period, historic data from their hospitals would be a second option to serve as a referent.
We could not find any similar sized company that was willing to share data. We nonetheless will keep on asking or monitoring literature and hope to get lucky enough to find a match for a comparison.
3) Perhaps the authors can contrast their analysis with other papers in the literature that have addresses similar hospital bed bottlenecks, such as during a natural disaster, war, influenza outbreak, etc.
It was possible to address this point in this part: lines 85-90: Spanish case studies [12 ; 13] reported their adopted strategies to face the outbreaks and the initial shock, contributing to the enrichment of the knowledge on the lessons learnt. Even if the hospitals’ robustness was smaller, those organizations put in place interven-tions (e.g., the use of indicators to catch the dynamicity of the organizations’ responses, the modeling of bed occupancy, etc.) that were able to exceed the structural limits of the hospitals.
4) A nice strength of the paper is it simplicity and readability. On the other hand, the manuscript noticeably lacks any inferential statistics and scientific rigor beyond basic tabulation by month. Some graphic visualization would be a plus.
We did not make any inference on the data we collected as a choice, as that will be the material for another article we are currently working on and that takes into consideration a larger timespan. We did not find the data we considered now as sufficient to infer, being them scarce.
we sincerely hope to see our paper published soon.
Round 2
Reviewer 1 Report
Dear Author(s)
Thanks for your review. I don´t have anything, more, to suggest to review.
Good luck in publishing your paper.
Best regards.
Reviewer 2 Report
Dear Authors,
Thank you very much for the revision.
I have no further comments.
Best Regards,